

# A novel simple immunoassay for quantification of blood anti-NMDAR1 autoantibodies

Melonie Vaughn[1], Susan Powell[1,2,3], Victoria Risbrough[1,3,4] and Xianjin Zhou[1,2,3]

[1] Department of Psychiatry, University of California, San Diego, La Jolla, CA, United States of America
[2] VA Mental Illness Research and Clinical Core, San Diego, CA, United States of America
[3] VA Research Service, San Diego, CA, United States of America
[4] VA Center of Excellence for Stress and Mental Health, San Diego, CA, United States of America

## ABSTRACT

Low titers of blood circulating anti-NMDAR1 autoantibodies have been reported in a significant subset of the general human population. Currently, immunohistochemical staining and cell-based assays are the standard methods for their detection and semi-quantification. However, detection and quantification of these low titers of blood circulating anti-NMDAR1 autoantibodies are problematic because of high non-specific background. Development of a new method to more accurately quantify these low titers of blood anti-NMDAR1 autoantibodies will facilitate studies on their potential impacts on psychiatric symptoms and cognition. We previously reported a robust production of anti-NMDAR1 autoantibodies against the ligand binding domain of NMDAR1. As a proof of principle, we report the development of a novel simple immunoassay for quantification of cross-species blood anti-NMDAR1 autoantibodies and its validation with immunohistochemistry and cell-based assays in both humans and mice. Specificity of our quantification was also investigated.

## INTRODUCTION

High titers of anti-NMDAR1 autoantibodies in the brain can cause anti-NMDAR1 encephalitis, a rare disease that displays a variety of psychiatric symptoms and neurological symptoms (*Dalmau, 2016*). Detection and semi-quantification of anti-NMDAR1 autoantibodies for investigation and diagnosis are mostly performed *via* immunohistochemical staining of rodent brains or the cell-based assay (*Gresa-Arribas et al., 2014*). However, we are interested in whether low titers of blood anti-NMDAR1 autoantibodies may impact cognitive functions and psychiatric symptoms since low titers of blood anti-NMDAR1 autoantibodies have been reported in the general human population and psychiatric patients (*Castillo-Gomez et al., 2017*; *Hammer et al., 2014*; *Jezequel et al., 2017*; *Pan et al., 2019*) and in mice (*Yue et al., 2021*). The cell-based assays were used for semi-quantification of low titers of human blood circulating anti-NMDAR1

Corresponding author
Xianjin Zhou, xzhou@ucsd.edu

autoantibodies. We previously found that detection of low titers of human blood anti-NMDAR1 autoantibodies using the cell-based assays suffers from high non-specific background (*Zhou, 2021a*). Additionally, these semi-quantitative methods are subjective in nature. Therefore, developing an objective method to quantify low titers of blood anti-NMDAR1 autoantibodies is important to advance our understanding of their potential impacts on psychiatric symptoms and cognition. We previously reported a GFP-based quick immunoassay to detect antibodies (*Yue et al., 2021*; *Zhou, 2021b*) and anti-NMDAR1 autoantibodies (*Yue et al., 2021*) that were robustly produced against the ligand binding domain (LBD) of NMDAR1 in mice. NMDAR1 LBD has been shown to be correctly folded after solubilization of over-expressed LBD from *E. coli* (*Furukawa & Gouaux, 2003*). As a proof of principle, we report the development of a *Gaussia* luciferase-based immunoassay to quantify the levels of blood anti-NMDAR1 autoantibodies between mice and humans as well as the validation of the luciferase-based immunoassay with the GFP-based assay, immunohistochemistry, and cell-based assays. Portions of the Introduction, Materials and methods, Results, and Discussion were previously published as part of a preprint (*Vaughn et al., 2024*).

## MATERIALS AND METHODS

### Expression of NMDAR1-luciferase fusion protein

The human NMDAR1 protein sequence was from NCBI reference sequence database (Accession NP_015566.1). The ligand binding domain (LBD) of NMDAR1 was synthesized with a linker GSGSG according to literature (*Furukawa & Gouaux, 2003*) and cloned into pET-21d vector. BL21(DE3)pLysS competent *E. coli* cells were purchased from EMD (cat. 70236-3) for transformation of the plasmids. The fusion proteins of the NMDAR1 LBD with either GFP or *Gaussia* luciferase (GLUC) were expressed in *E. coli* and purified with HisPur Ni-NTA resins (*Zhou, 2021b*) before solubilized as described (*Waldo et al., 1999*) and folded as described (*Furukawa & Gouaux, 2003*). Successful folding of the probe was examined with either GFP fluorescence or activities of *Gaussia* luciferase. *Gaussia* luciferase (GLUC) was also produced as a control. These protein probes are produced in Xianjin Zhou laboratory and are available upon request.

### Human plasma samples

Human plasma samples were leveraged from an existing sample repository with approval from the San Diego Veterans Affairs Healthcare Services (IRB Protocol H180112). Plasma samples were collected with BD Vacutainer™ Plastic Blood Collection Tubes with Lithium Heparin after written consent was obtained. Samples were stored in −80 freezers until use. We used 143 blood plasma samples from young healthy male participants (average age 22.8 years, range 19–23). At the time of blood collection these participants endorsed no major physical or mental health disorders.

### Mouse strain and active immunization

C57BL/6J mice were purchased from Jackson Labs (Bar Harbor, ME) at 8 weeks-old. Mice were housed in standard cages (28.4 × 18.4× 12.5 cm) in an individually ventilated

caging system with corncob bedding and fed with Teklad (Envigo). Nesting materials were provided as enrichment for each cage. All mice were housed in groups of four in an AAALAC-approved UCSD facility and maintained on a reversed light/dark cycle, lights on from 7 pm to 7 am. The studies were approved by University of California San Diego (S04190M) and San Diego Veterans Affairs Healthcare Services Animal Care and Use Committee prior to the onset of the experiments. Mice were maintained in American Association for Accreditation of Laboratory Animal Care approved animal facilities at the local Veteran's Administration Hospital or UCSD campus. These facilities meet all Federal and State requirements for animal care. Active immunization was conducted after a week acclimation and produced anti-NMDAR1 autoantibodies in all 20 mice immunized with the P2 antigen, and these mice displayed spatial working memory deficits with a large effect size and a 0.9 to 1 statistical power in our previous studies (*Yue et al., 2021*). For the immunization, P2 peptide antigens of mouse NMDAR1 were synthesized by *Biomatik*. Mice were randomly assigned in excel form for immunization with either the control CFA (20 mice, 10 mice/sex) or the CFA plus the P2 antigen (20 mice, 10 mice/sex). Neither analgesia nor anesthesia were used since immunization is transient with mild pain. In brief, the P2 peptide was dissolved in PBS at a concentration of four mg/ml and mixed with an equal volume of Complete Freund's Adjuvant to generate a thick emulsion. Mice are monitored 2–6 h post immunization and once daily for general health after that. If any of the following are observed (abscesses or ulcers, abnormal eating and/or drinking, loss of weight [>20%], non-aggressive vocalizations when handled, lack of movement [*i.e.*, lethargy], squinting of the eyes, and lack of grooming/severely ruffled fur), mice are assessed, treated appropriately, and euthanized if necessary. Euthanasia prior to the planned end of the experiment was not needed in these studies. Two months after immunization, a few microliters of blood were collected from mouse tail vein for analysis of anti-NMDAR1 autoantibodies. Blood collection was blindly conducted to the treatment of each mouse. All mice used in these experiments were euthanized at the conclusion of the experiment. Mice were euthanized by rapid asphyxiation from carbon dioxide gas in non-crowded chambers. After death by asphyxiation, mice are cervically dislocated to assure death.

## Immunohistochemistry

Immunohistochemistry (IHC) was conducted as previously described (*Ji et al., 2013*; *Kim et al., 2012*) to semi-quantify the levels of blood anti-NMDAR1 autoantibodies in individual mice. Wildtype mouse brain paraffin sections were used for IHC analysis with a dilution of mouse serum at 1:200 with antibody diluent solution (S080983-2; Dako, Glostrup, Denmark). Mouse anti-NMDAR1 monoclonal antibody (cat. 556308; BD Biosciences, Franklin Lakes, NJ, USA) was used as the positive control with a dilution at 1:40,000. ImmPRESS peroxidase-micropolymer conjugated horse anti-mouse IgG (H + L) (MP-7402; Vector Labs, Newark, CA, USA) was used as the secondary antibody. Biotinylated goat anti-human IgG (H + L) secondary antibody was used for detection of human plasma anti-NMDAR1 autoantibodies. Chromogenic reaction was conducted with ImmPACT NovaRED Peroxidase Substrate (SK-4805; Vector Labs, Newark, CA, USA). Slides were mounted with Cytoseal 60 mounting medium (8310-16; Richard

Allan Scientific, Kalamazoo, MI, USA). *Image J* was used to measure differential optical intensities between hippocampal CA1 *st oriens* and *corpus callosum* to semi-quantify the levels of anti-NMDAR1 autoantibodies.

### Immunoassay

Due to insufficient serum leftover after immunohistochemistry from four mice (two control, two P2 mice), we conducted the immunoassay with triplications for the remaining 18 control mice and 18 mice immunized with the P2 antigen. The immunoassay was blindly conducted to the treatment of each mouse. Protein A/G/L was purchased from Novus Biologicals (NBP2-34985) and diluted to one ug/ul with antibody diluent solution (DAKO, S080983-2). Per functionality report (see Novus Biologicals, NBP2-34985), purchased protein A/G/L has 12 established antibody binding sites. IgG, the most abundant antibody isotype in humans, ranges from 7.5 to 22 mg/mL in human blood serum. With one µg of AGL binding to over 10 µg of IgG, we opted to use two µg of AGL for greater pipetting ease/accuracy after seeing no difference in relative autoantibodies levels between with variations in protocol. Two microliters of mouse or human serum/plasma was incubated with NMDAR1-GLUC and two ul of protein A/G/L in 1X PBS, 0.25M NaCl, 1% Triton X100. The mixture was then incubated for 1.5 h at room temperature. After adding 200 ul of 1XPBS, 0.1% Tween 20 for washing, the pellet was collected by centrifugation at 3,220 rpm (1,889 x g, Eppendorf, Centrifuge 5810R) for 10 min. The pellet was washed again with 200 ul of 1XPBS. After the washing, the pellet was suspended in 10 ul of 1XPBS. 20 ul of *Gaussia* luciferase substrate (ThermoFisher, cat. 16160; Pierce™ Gaussia Luciferase Glow Assay Kit) was then added to the suspended pellet. *Gaussia* luciferase activities were stabilized for 10 min at room temperature. Luciferase activities (RLU) were measured with Greiner 96-well Flat Bottom Black Polystyrene plate (Cat. No.: 655097) on Tecan infinite 200Pro. The levels of anti-NMDAR1 P2 autoantibodies were quantified for each mouse. Student's *t*-test was used for statistical analysis of anti-NMDAR1 P2 autoantibodies between the control mice and the mice immunized with the P2 antigen.

### Cell-based assay

Immunofluorescence analysis of mouse and human anti-NMDAR1 autoantibodies were conducted using Euroimmun BIOCHIP with a positive control of human anti-NMDAR1 autoantibody as described in our previous studies (*Yue et al., 2021*). Mouse serum and human plasma were diluted at 1:10 and 1:50, respectively with DAKO antibody diluent solution. Goat anti-Human IgG (H+L) Fluorescein (FI-3000; Vector Labs, Newark, CA, USA) and goat anti-mouse IgG (H+L) Alexa Fluor 568 (A11004; Invitrogen, Waltham, MA, USA) were diluted at 1:1,000 as the secondary antibodies for immunofluorescence staining. Flurorescence staining was examined using microscope EVOS FL (Thermo Fisher Scientific, Waltham, MA, USA).

## RESULTS

We previously developed the One-Step GFP-based assay that enables instant visualization of antibody using protein AGL to aggregate all isotypes of antibodies during antibody binding
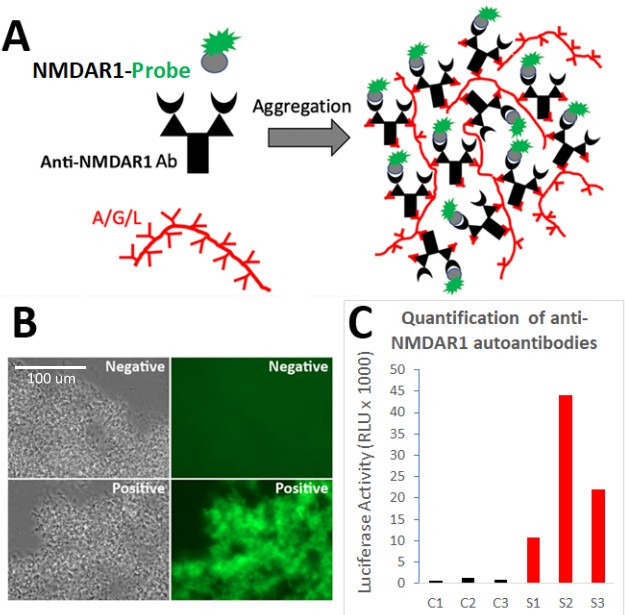

**Figure 1 Development of a luciferase-based immunoassay to quantify blood anti-NMDAR1 autoantibodies.** (A) Strategy for the one-step immunoassay (*Yue et al., 2021*; *Zhou, 2021b*). (B) One-step immunoassay when the probe is GFP. Blood from negative control mice and mice carrying the anti-NMDAR1 P2 antibodies are shown. All isotypes of blood antibodies are aggregated by protein A/G/L (gray image). Anti-NMDAR1 P2 autoantibodies within the aggregate from a positive mouse bind NMDAR1-GFP and emit strong green fluorescence. Aggregated antibodies from negative control mouse blood showed little background. Scale bar: 100 μm. (C) Luciferase-based immunoassay when the GFP probe is replaced by *Gaussia* luciferase. Quantification of the levels of blood anti-NMDAR1 autoantibodies in positive mice (S1–S3) and negative control mice (C1–C3).

GFP-labelled antigens (*Yue et al., 2021*; *Zhou, 2021b*). To quantify the levels of blood anti-NMDAR1 autoantibodies recognizing NMDAR1 ligand binding domain, we fused the ligand binding domain of human NMDAR1 with *Gaussia* luciferase (Fig. 1A). After folding of the probes (*Furukawa & Gouaux, 2003*), mouse anti-NMDAR1 autoantibodies generated in our previous studies (*Yue et al., 2021*) were detected with either the NMDAR1-GFP (Fig. 1B) or quantified with the NMDAR1 fused *Gaussia* luciferase (Fig. 1C). We observed negligible non-specific background in either assay, supporting feasibility for developing a luciferase-based assay to quantify the levels of the blood anti-NMDAR1 autoantibodies.

To validate the luciferase-based immunoassay, we conducted correlation studies of mouse blood anti-NMDAR1 autoantibodies with immunohistochemistry that was published in our previous studies (*Yue et al., 2021*). Immunized mice developed anti-NMDAR1 autoantibodies (Fig. 2A) (Data S1). For semi-quantification by immunohistochemistry, optical intensities of CA1 staining (CA1 *st oriens*), after subtracting the background intensity from *corpus callosum*, were used to quantify the relative levels of the blood anti-NMDAR1 autoantibodies for individual mice. Mice immunized with the NMDAR1 P2 antigen developed significantly higher levels of anti-NMDAR1 autoantibodies than the control mice immunized with Complete Freund Adjuvant (CFA) alone using the

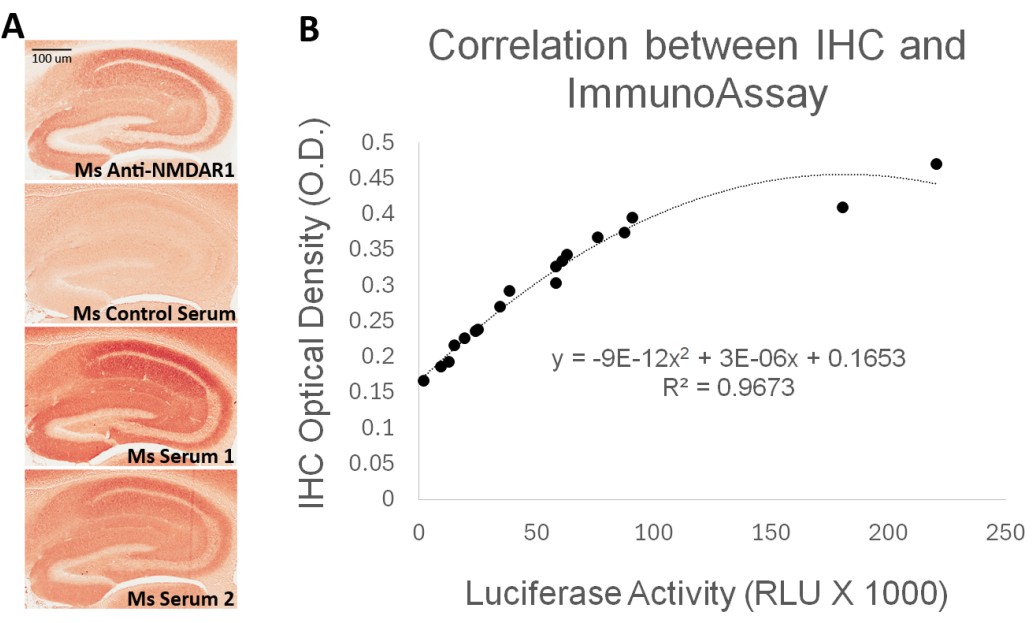

**Figure 2** **Correlation analysis between quantification by the luciferase-based immunoassay and semi-quantification by immunohistochemistry (IHC).** (A) Wildtype mouse paraffin hippocampal sections were used for immunohistochemical analysis of anti-NMDAR1 autoantibodies. Sera of immunized mice were diluted at 1:200 for the IHC analysis. Positive control (Ms Anti-NMDAR1): a commercial mouse anti-NMDAR1 monoclonal antibody (1:40,000 dilution). Negative control (Ms Control Serum): mouse immunized with CFA only. Test mice (Ms Serum 1 and 2): representatives from 18 mice immunized with CFA and NMDAR1 P2 antigens. Optical intensities of hippocampal CA1 *st oriens* and *corpus callosum* were quantified with Image J and their differences were used as the relative levels of anti-NMDAR1 autoantibodies in individual mice. Scale bar: 100 μm. (B) Anti-NMDAR1 autoantibodies from 18 mouse serum samples were quantified by the luciferase-based immunoassay. A correlation between the IHC semi-quantification and the luciferase quantification of blood anti-NMDAR1 autoantibodies was shown for these 18 serum samples. A strong correlation ($R^2 = 0.97$) was observed between these two methods.

semi-quantification method (Student's $t$-test, $p = 4.31988E–11$). The levels of the blood anti-NMDAR1 autoantibodies of individual mice were also quantified using the luciferase-based immunoassay. All raw quantification data are in Data S2. A high correlation ($R^2 = 0.97$) was observed between the levels of blood anti-NMDAR1 autoantibodies quantified with immunohistochemistry and the luciferase-based assay (Fig. 2B). However, ceiling effects seemed to impact the highest levels of anti-NMDAR1 autoantibodies in the immunohistochemistry assay. This finding suggests that more serum dilutions are needed to accurately quantify the high levels of anti-NMDAR1 autoantibodies to avoid color saturation in chromogenic immunohistochemistry. In contrast, activities of *Gaussia* luciferase have a linear range from a thousand to a million relative light units (RLU). Such ceiling effects are not present in the luciferase-based assay; and no dilutions are needed to achieve accurate quantification of high or low levels of the anti-NMDAR1 autoantibodies.

Since the protein A/G/L binds all isotypes of antibodies across mammals, we conducted comparative analyses of the levels of blood circulating anti-NMDAR1 autoantibodies between mice and humans using our luciferase-based immunoassay (Fig. 3). All mice

immunized with NMDAR1 P2 peptide antigen (red dots) except one have higher levels of anti-NMDAR1 autoantibodies than the background level from the negative control mice (black dots). Significant higher levels ($p < 0.001$) of anti-NMDAR1 P2 autoantibodies were detected in mice immunized with CFA and P2 than in mice immunized with CFA only. Human plasma (blue dots) samples were taken from 143 young healthy males with ages ranging from 19 to 23 years old. All raw data are in Data S3. A higher background level of natural anti-NMDAR1 autoantibodies was observed in human plasma than in mice. The physiological significance of this background difference is unknown, and the difference may be partly attributed to higher concentrations of overall blood antibodies in humans than in mice. The interquartile range (IQR) was used to identify statistical outliers (Q3+1.5*IQR); and more than 10% of the human subjects were the outliers with higher levels of anti-NMDAR1 autoantibodies, supporting previously reported qualitative findings that a significant subset of the general human population carries anti-NMDAR1 autoantibodies in blood.

Binding of plasma antibodies to the *Gaussia* luciferase may potentially be detected as false positives of anti-NMDAR1 autoantibodies by the NMDAR1-*Gaussia* luciferase probe. Therefore, we re-examined the quantified samples that carry high levels of anti-NMDAR1 autoantibodies with a different NMDAR1-GFP fusion probe. The top 20 plasma samples carrying the highest levels of anti-NMDAR1 autoantibodies were also positive for the GFP-based assay, indicating that the presence of the antibodies cross-reacting with *Gaussia* luciferase was minimal in the 143 human plasma samples. Out of these 143 samples, the human plasma carrying the highest levels of anti-NMDAR1 autoantibodies emit strong green fluorescence after protein A/G/L aggregation in contrast to the plasma with basal levels of anti-NMDAR1 autoantibodies (Fig. 4). Different fusion probes with either the GFP or the luciferase provide a cross validation for analysis of anti-NMDAR1 autoantibodies. We did not find any human plasma significantly binding the luciferase after screening >300 human plasma samples including these 143 plasma samples.

The anti-NMDAR1 autoantibody-positive plasma identified from the luciferase-based assay were also verified using immunohistochemical staining of mouse hippocampus, the assay commonly used for the detection of anti-NMDAR1 autoantibodies. NMDAR1 staining signal in hippocampus was stronger from human plasma 2 carrying the highest level of anti-NMDAR1 autoantibodies than from human plasma 1 carrying a basal level of the anti-NMDAR1 autoantibodies (Fig. 5). In comparison to the little NMDAR1 staining by control mouse serum (see previous Fig. 2), there is some weak NMDAR1 staining from human plasma on mouse hippocampus. This finding is consistent with our cross-species quantification findings that human plasma has a higher basal level of anti-NMDAR1 autoantibody binding activity than mouse serum.

The anti-NMDAR1 autoantibody-positive plasma identified from the luciferase-based assay was further verified using the cell-based assay that is also commonly used for detection of anti-NMDAR1 autoantibodies. As expected, immunocytochemical staining by human anti-NMDAR1 autoantibodies and mouse anti-NMDAR1 P2 autoantibodies demonstrated complete co-localization (top panel, Fig. 6). There are some differential staining intensities between the two different anti-NMDAR1 antibodies. It is possible that the accessibility

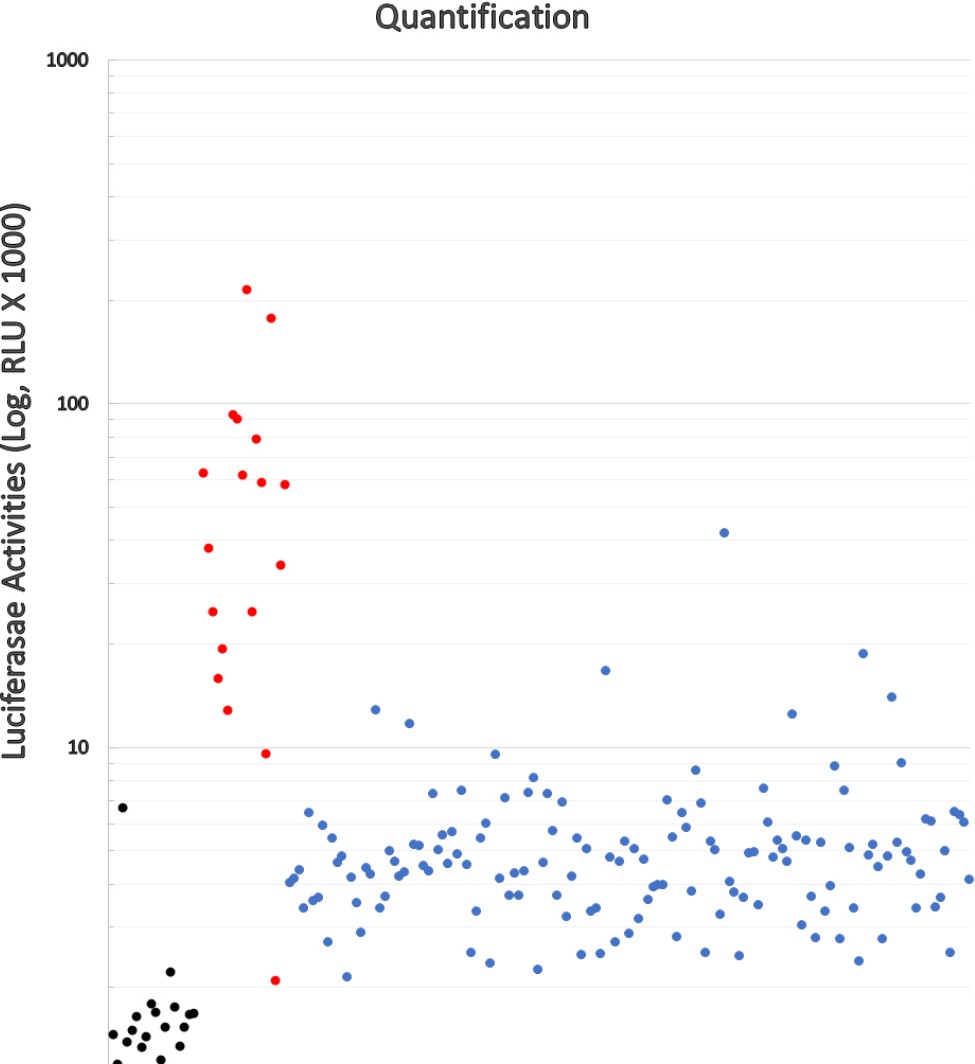

**Figure 3** **Cross-species quantification and direct comparison of blood anti-NMDAR1 autoantibodies between mice and humans.** Anti-NMDAR1 autoantibodies were quantified in the blood of 143 human subjects (blue dots), negative control mice (black dots), and mice immunized with NMDAR1 P2 antigen (red dots). Activities of *Gaussia* luciferase (RLU, X1000) are used as the levels of anti-NMDAR1 autoantibodies (Log10 scale). A significant portion of the general human population carries natural anti-NMDAR1 autoantibodies in blood.

of different antigenic epitopes of NMDAR1 proteins may contribute to these differences. To detect lower levels of plasma anti-NMDAR1 autoantibodies found in the general human population, human plasma requires less dilution. However, less diluted plasma

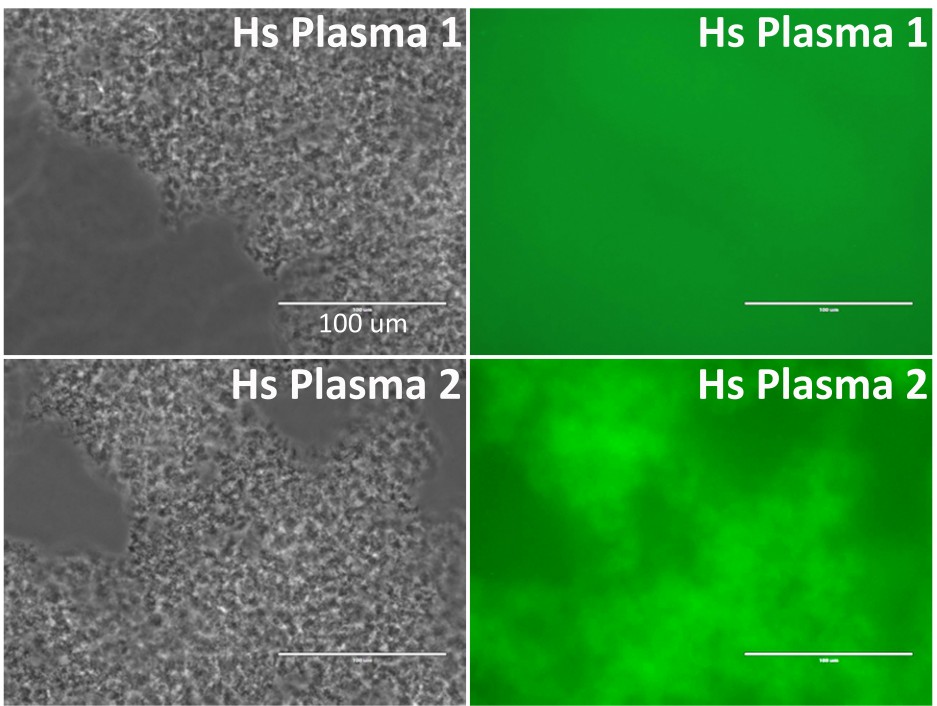

**Figure 4** **Validation of anti-NMDAR1 autoantibodies with a different NMDAR1-GFP probe.** A representative human plasma (Hs Plasma 1) carrying a basal level of anti-NMDAR1 autoantibodies. Human plasma 2 (Hs Plasma 2) carries the highest level of natural anti-NMDAR1 autoantibodies in the 143 human plasma samples. To validate the luciferase-based assay, a GFP-based immunoassay was conducted. Blood antibodies are aggregated by protein A/G/L (gray image). Anti-NMDAR1 autoantibodies within the aggregate from Hs Plasma 2 bind NMDAR1-GFP and emit green fluorescence. Aggregated antibodies from Hs Plasma 1 showed very weak signal. Scale bar: 100 µm.

generated higher non-specific staining background in the cell-based assays (bottom panel, Fig. 6). Anti-NMDAR1 autoantibody staining can only be recognized with the help of co-immunocytochemical staining with known anti-NMDAR1 antibodies.

Finally, we investigated how much non-specific binding contributes to the quantification of anti-NMDAR1 autoantibodies in both mice and humans. The control GLUC probe without NMDAR1 fusion was used to assess the levels of both non-specific binding and GLUC-binding activities. We randomly selected human plasma samples that carry different levels of anti-NMDAR1 autoantibodies. With the same amount of input GLUC or NMDAR1-GLUC luciferase activities, we found a very low level of GLUC-binding antibodies across mouse serum and human plasma samples regardless of the levels of anti-NMDAR1 autoantibodies measured by the NMDAR1-GLUC probe (Fig. 7). All raw data are in Data S4. This data suggested that antibodies binding NMDAR1 ligand binding domain were specifically measured for individual plasma samples.

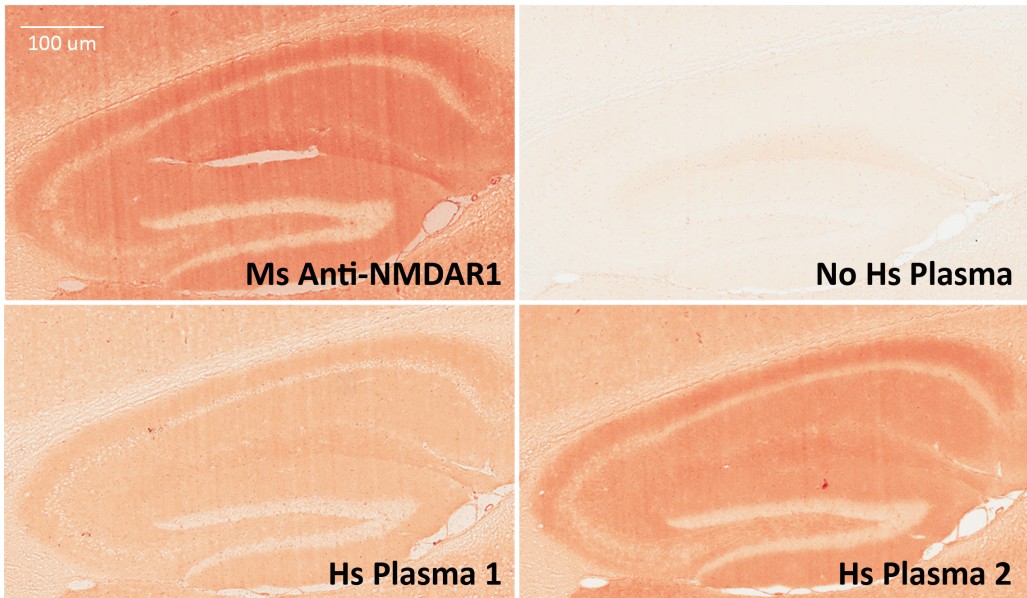

**Figure 5** **Validation of anti-NMDAR1 autoantibodies with immunohistochemistry.** Mouse hippocampal paraffin sections were used for detection of human anti-NMDAR1 autoantibodies. Human plasma was diluted at 1:500 for the IHC analysis. Hippocampal staining with a commercial mouse anti-NMDAR1 monoclonal antibody (1:40,000 dilution) was used as the positive control. A typical hippocampal NMDAR1 staining pattern was observed. Compared to weak signal from Hs Plasma 1 carrying a basal level of anti-NMDAR1 autoantibodies, strong signal was observed from Hs Plasma 2 carrying the highest level of anti-NMDAR1 autoantibodies. A hippocampal section was also included as a negative control without addition of primary antibodies or human plasma but only biotinylated goat anti-human IgG (H + L) secondary antibodies. Scale bar: 100 μm.

## DISCUSSION

A significant portion of the general human population carry anti-NMDAR1 autoantibodies in their blood circulation, but their cognitive effects remain unclear (*Zhou, 2021a*). In addition, there have been many studies on the potential effects of blood circulating anti-NMDAR1 autoantibodies on psychiatric disorders (*Castillo-Gomez et al., 2017*; *Hammer et al., 2014*; *Jezequel et al., 2017*; *Pan et al., 2019*), cognitive functions (*Yue et al., 2021*), and neurodegenerative diseases (*Hopfner et al., 2019*). All these studies used cell-based assays to detect human blood anti-NMDAR1 autoantibodies and semi-quantify their levels after a series of dilutions. High non-specific background staining is inevitable for detection of such lower levels of blood anti-NMDAR1 autoantibodies using this assay. The high background of the cell-based assay is mainly generated from various host cell proteins reacting with different antibodies in human plasma/serum. To minimize this non-specific background, we developed a novel immunoassay for the detection and quantification of blood anti-NMDAR1 autoantibodies, based on our previously reported one-step immunoassay. In this assay, only antibodies binding NMDAR1 LBD are quantified.

Cell-based assay and immunohistochemical staining are currently the most used methods to quantify blood anti-NMDAR1 autoantibodies. A series of dilutions of serum/plasma

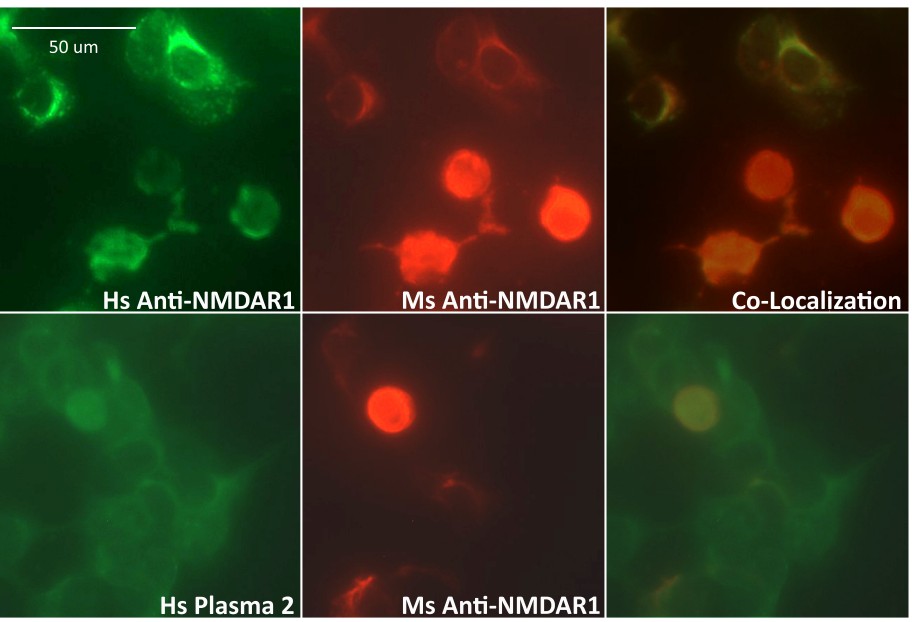

**Human Plasma: 1:50 dilution**

**Figure 6** **Validation of anti-NMDAR1 autoantibodies with cell-based assay.** Human NMDAR1 proteins were expressed on HEK293 cells on BIOCHIPs purchased from Euroimmun. Both anti-Human NMDAR1 autoantibody (Euroimmun) and the mouse anti-NMDAR1 serum (diluted at 1:10) against the NMDAR1 P2 peptide antigens recognize the NMDAR1 proteins on HEK293 cells. The staining between the anti-Human NMDAR1 and mouse anti-NMDAR1 serum was co-localized in cell-based assays (top panel). Both human plasma 2 (diluted at 1:50) and mouse anti-NMDAR1 serum (diluted at 1:10) were used for co-immunofluorescence staining on BIOCHIPS. A high non-specific background was observed for human plasma. Co-localization of the staining between human plasma 2 and mouse anti-NMDAR1 serum (bottom panel), suggesting the presence of human natural anti-NMDAR1 autoantibodies. Scale bar: 50 μm.

are needed for such assays, and detection is subjective. Compared to these labor intensive and less accurate semi-quantification methods, our luciferase-based immunoassay offers an efficient and objective quantification that is particularly helpful for analysis of low levels of blood anti-NMDAR1 autoantibodies in psychiatric patients and the general human population. In addition, our luciferase-based immunoassay can be adapted for high throughput quantification of anti-NMDAR1 autoantibodies when conducting the assay on a large scale. Finally, our quantification method does not require a secondary antibody and protein A/G/L binds antibodies from different mammals, enabling quantification and direct comparison of the levels of blood anti-NMDAR1 autoantibodies across mammalian species.

A potential complication of our luciferase-based immunoassay could arise from antibodies recognizing the *Gaussia* luciferase part of the fusion protein rather than the NMDAR1 antigen part. However, we found an insignificant amount of these non-specific

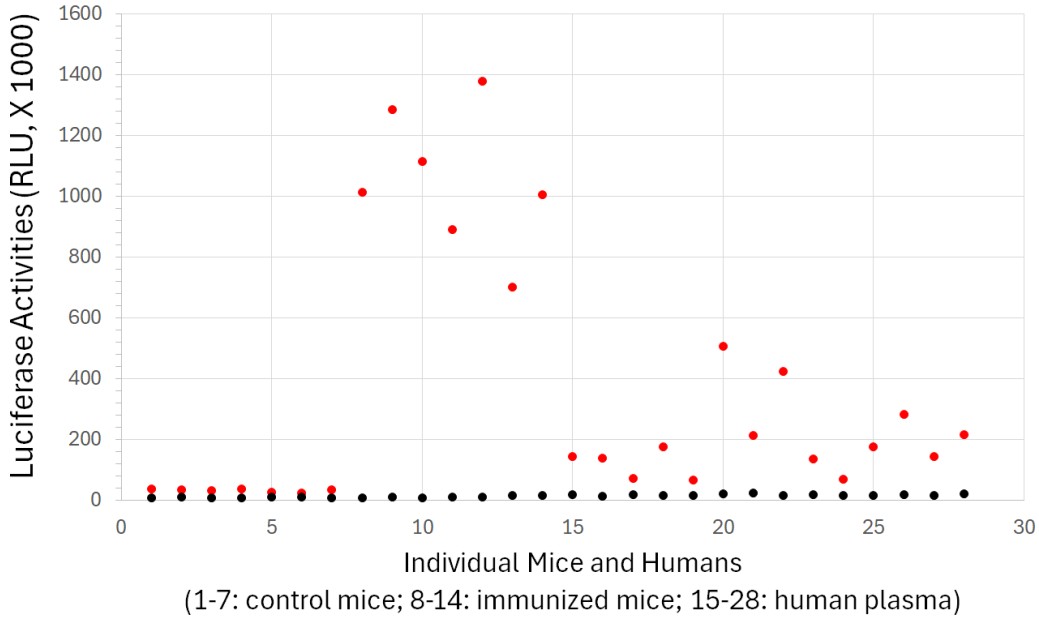

**Figure 7** **Little contribution from non-specific antibody binding to quantification of anti-NMDAR1 autoantibodies.** Specificity of the quantification of anti-NMDAR1 autoantibodies were validated in both immunized mice and human subjects using both GLUC and NR1-GLUC probes. The same amount of input luciferase activities from either GLUC or NR1-GLUC probe was used for quantification using the same protein AGL mixture to assess both non-specific binding and specific anti-NMDAR1 antibody binding across mouse and human. Control wildtype mice (1–7); immunized mice carrying anti-NMDAR1 autoantibodies (8–14). Randomly selected human subjects carrying different levels of anti-NMDAR1 autoantibodies (15–28).

binding and GLUC-binding activities in the quantification, particularly in the subjects who carry higher levels of anti-NMDAR1 autoantibodies.

Our studies provided proof-of-concept for the development of a simple method to quantify blood anti-NMDAR1 autoantibodies. Different antigens can however be fused with *Gaussia* luciferase for detection of their antibodies after aggregation. NMDAR1 LBD has been shown to be correctly folded after solubilization of over-expressed LBD from *E. coli* (*Furukawa & Gouaux, 2003*). The NMDAR1 LBD-luciferase fusion proteins can be abundantly produced in *E. coli*, which makes the quantification efficient and cost effective. However, fusion proteins produced from *E. coli* lack post-translational modifications. For detection of antibodies recognizing these modifications, the NMDAR1-luciferase fusion proteins need to be produced from mammalian cells. Since protein A/G/L aggregates all isotypes of antibodies, our method cannot directly detect different isotypes of anti-NMDAR1 autoantibodies individually. However, if streptavidin and biotinylated antibodies for a specific isotype of antibodies are used to aggregate antibodies rather than protein A/G/L, it will be feasible to quantify each isotype of blood anti-NMDAR1 autoantibodies.

A limitation of our current study is lack of a positive control of human anti-NMDAR1 autoantibodies against the ligand binding domain of NMDAR1. Most anti-NMDAR1 autoantibodies from patients with anti-NMDAR1 encephalitis bind the amino terminal domain, but not the ligand binding domain as tested here (*Dalmau et al., 2008*). However, it is important to note that our study was not intended to develop an immunoassay for diagnosis of anti-NMDAR1 encephalitis, but rather a prototype of a new immunoassay to quantify plasma anti-NMDAR1 autoantibodies across different species. Another limitation is that since all human subjects in our study are young males, effects of different age ranges and genders are not investigated. Finally, effects of the plasma anti-NMDAR1 autoantibodies on human cognitive functions and diseases remain to be investigated in the future.

### Funding
This work was supported by the grant R01NS135620 (Xianjin Zhou, Melonie Vaughn, and Victoria Risbrough). Melonie Vaughn is also supported by a National Science Foundation Graduate Fellowship. Victoria Risbrough is supported by a BLR&D Veterans Affairs Research Career Scientist Award and the VA Center of Excellence for Stress and Mental Health. Susan Powell is supported by a VA Merit Award. The funders had no role in study design, data collection and analysis, decision to publish, or preparation of the manuscript.

### Grant Disclosures
The following grant information was disclosed by the authors:
National Science Foundation Graduate Fellowship: R01NS135620.
BLR&D Veterans Affairs Research Career Scientist Award.
VA Center of Excellence for Stress and Mental Health.
VA Merit Award.

### Competing Interests
Xianjin Zhou is the inventor for UCSD Patent: U.S. Appln No: 63/088,025: METHODS FOR DETECTING ANTIBODIES.

### Author Contributions
- Melonie Vaughn performed the experiments, analyzed the data, prepared figures and/or tables, and approved the final draft.
- Susan Powell performed the experiments, analyzed the data, prepared figures and/or tables, and approved the final draft.
- Victoria Risbrough analyzed the data, prepared figures and/or tables, and approved the final draft.
- Xianjin Zhou conceived and designed the experiments, performed the experiments, analyzed the data, prepared figures and/or tables, authored or reviewed drafts of the article, and approved the final draft.

## Human Ethics

The following information was supplied relating to ethical approvals (i.e., approving body and any reference numbers):

San Diego Veterans Affairs Healthcare Services

## Animal Ethics

The following information was supplied relating to ethical approvals (i.e., approving body and any reference numbers):

UCSD Animal Care Program

## Data Availability

The raw measurements are available in the Supplementary Files.

## Supplemental Information

Supplemental information for this article can be found online at http://dx.doi.org/10.7717/peerj.19212#supplemental-information.

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
