# Peer review of "A novel simple immunoassay for quantification of blood anti-NMDAR1 autoantibodies"

_PeerJ, doi:10.7717/peerj.19212_

## Round 0.1 · original submission · Major Revisions

The study demonstrates that peptide-specific antibodies can be generated in rodents and detected using the authors' assay; however, claims about its diagnostic utility for NMDAR encephalitis lack sufficient support. To strengthen the manuscript, the authors should focus on the animal immunization findings and either remove speculative diagnostic claims or provide additional experimental data to substantiate the assay's potential diagnostic applications.

Please address all reviewers' comments in the revised manuscript and include a detailed point-by-point response.

Reviewer 1 ·

Basic reporting

This study reported the development of a novel simple immunoassay for
quantiƻcation of cross-species blood anti-NMDAR1 autoantibodies.

Experimental design

1. The human NMDAR1 protein sequence from the NCBI reference sequence database should be NP_015566.1, not NP_0155661.1.
2. All the images are required to have a uniform and clear scale bar.
3. For the results in Figure 2A, please provide the grayscale statistical results of the images.
4. Figure 5 only presents the results of two human serum samples. Could you provide the results of other samples? It would be best to make grayscale statistical charts for each group as well.
5. In Figure 6, can the first 20 high-value samples screened out all be confirmed by the cell-based assay? If not, please elaborate on the verification status of these samples and provide the corresponding image data. Besides, has the pathogenicity of these 20 high-value serum samples been studied?

Validity of the findings

This study validation with immunohistochemistry and cell-based assays in both humans and mice.

Additional comments

What is the threshold for differentiating between serum - negative and - positive?

·

Basic reporting

1. In the introduction, it would be better if you include a line that explains the rationale of choosing NMDAR 1 subunit and P2 antigen for immunisation instead of other subtypes.

2. The conclusions were not presented in a separate section. Instead, those seem to be included in the discussion.

Experimental design

3. In the method section, could you please provide more details/ test algorithm for human testing? It is briefly mentioned in the introduction and in human plasma samples.

4. In the results, could you clarify the cutoff for determining "high levels"? Was it only the 10% of outliers?

Validity of the findings

No comment

Reviewer 3 ·

Basic reporting

The basic reporting of what was done in the study is reasonable.

Some minor points, the authors do non mention what the P2 antigen is (I appreciate it is in one of their previous publications). I don't see where the authors have shown the capacity of the Protein A/G/L beads or at least demonstrate that it captures all the antibodies in 2 ul mouse and human serum. When reporting centrifugation it is standard to report as x gravity. RPM does not inform on the force as it depends on the distance from the centre of the rotor. The strong positive human sample in Figure 6 does not co-localise with the commercial antibody, which itself appears to predominantly bind the nucleus of the HEK cells?

I do not see where the authors show any evidence that human patients with NR1 antibodies bind the LBD domain. The original publication from Dalmau et al in the lancet in 2008 suggest that antibodies bind the ATD and rarely the LBD. In 92/100 samples from patients with NMDAR antibodies the binding was lost when residues 25-380 were removed and in 8 it was significantly reduced.

Experimental design

The authors have generated two bacterially produced soluble NR1-LBD-GFP/Gluc molecules to detect antibodies in mice that were immunised with CFA and a P2 peptide, or just CFA alone, and compared with healthy male volunteers.

The conformation of the antigen is key to antibody detection in patients with NMDAR encephalitis as has been described in the literature over the last 20 years.
In this work the authors suggest that the presence of active GFP or luciferase is a reasonable surrogate for LBD conformation. I would argue that the authors provide no evidence for correct folding of the LBD in this paper. As the mice were immunised with a peptide, then native conformation is not required for binding. Additionally, looking at the crystal structure the chosen peptide does not appear on the surface of the protein, which would preclude patient antibodies binding to the native protein in vivo.

More recently patient derived monoclonal antibodies have been described that uniquely bind the ATD of the NR1 subunit (Michalski et al. Nature structural and molecular biology 2024; Wang et al. Nature structural and molecular biology 2024). We do not understand the relative frequency of ATD and LBD autoantibodies in patients with NMDAR antibodies. By leaving out the ATD in these assays we are missing patient antibodies and reducing test sensitivity. Additionally, work at the Mayo Clinic demonstrated that in protein microarray NR1 antibodies were not identified in serum samples from patients with high titre antibodies, suggesting that the native conformation is important for this test (McKeon et al N2 2023).

So the authors are generating antibodies to a peptide that can be precipitated by the construct that they have generated, and these antibodies bind parafin fixed rodent tissue by IHC. This is reasonable but it is a leap to suggest that this will be useful in detection of antibodies in patients with NMDAR encephalitis based on the current literature and no evidence in this paper.

Validity of the findings

There are no sera from patients with NMDAR encephalitis in the paper. There are no human disease controls in this study. The authors suggest that there is a background issue when sera from young healthy males are screened. I expect this background to be worse in people with an inflammatory disease.
A cohort of serum from patients with antibody mediated CNS immunity is required as a control for the assay. This should include a subgroup of samples at relapse to challenge the assay.
Previous attempts at immunoprecipitation of antibodies using GFP tagged AQP4 at the Mayo Clinic for patients with NMOSD showed that this assay was not sensitive and is not used clinically (Waters et al Neurology 2012). There is no evidence in this publication to suggest this would be better or equivalent to current technologies.
The animal work is not relevant for an assay to detect antibodies associated with human disease.

Additional comments

n/a

Reviewer 4 ·

Basic reporting

The authors present a clear methods paper describing a novel immunoassay for detecting NMDA receptor autoantibodies. Figures, raw data appear to be appropriate.

Minor edits:
Line 64: change "most" to "mostly"
Line 71: change "semi-quantitations" to "semi-quantitation"

Experimental design

Methods are well-described. A statement about availability of these custom reagents may be necessary to fulfill the requirement that this study can be replicated by an independent investigator.

While mouse samples adequately separate positive vs negative controls, there are no positive controls for the human samples. Are all human samples from subjects who are free of neurologic/psychologic symptoms? If so, this should be stated in the methods.

An ideal positive control in human plasma would be a patient with NMDA receptor encephalitis. As these are rare, looking at an increase in prevalence of auto-antibodies with age might provide more dynamic range (see Busse et al DOI: 10.1007/s00406-014-0493-9).

Validity of the findings

The conclusions support the assay development and validation in mouse. Lack of positive control in humans should be discussed, if additional positive control samples cannot be acquired. Authors should also note limitations of the human dataset (male only, limited age range).

Other factors would have to be evaluated to understand assay performance before using this assay as a diagnostic, e.g. impact of dehydration or other inflammatory condition, and parameters such as freeze-thaw, length of time in storage, etc.

Additional comments

It would be good to lay out next steps needed to properly validate the assay (both analytical and biological validation) at the end of the discussion. While these data describe a potentially useful new method, the data are preliminary.

---

## Round 0.2 · Minor Revisions

Address all remaining comments made by reviewers and submit revised manuscript along with point wise responses.

·

Basic reporting

No comment

Experimental design

"A potential complication of our luciferase-based immunoassay could arise from antibodies recognizing the Gaussia luciferase part of the fusion protein rather than the NMDAR1 antigen part. However, we found an insignificant amount of these non-specific binding and GLUC-binding activities in the quantification, particularly in the subjects who carry higher levels of anti-NMDAR1 autoantibodies."
1) Does this mean that there may be a significant quantification error/ higher proportion of nonspecific binding in low NMDAR titers or is the percentage of nonspecific binding even lower at low titers?
2) In the results section, it was mentioned as low binding regardless of the level of NMDAR1 & figure 7 shows the same.

Validity of the findings

No comment

Additional comments

No comments

Reviewer 4 ·

Basic reporting

The manuscript has been clarified on several points, and the authors have addressed primary concerns.

Experimental design

The manuscript has been clarified on several points, and the authors have addressed primary concerns.

Validity of the findings

The manuscript has been clarified on several points, and the authors have addressed primary concerns.

Additional comments

One minor comment... does the statement about the pre-print have to precede each section? I would recommend a single statement at the end of the introduction, and specify that text from multiple sections of this manuscript were included in a pre-print.

---

## Round 0.3 · accepted · Accept

Authors have addressed all of the reviewers' comments and manuscript is ready for the publication.

·

Basic reporting

no comment

Experimental design

no comment

Validity of the findings

no comment

Additional comments

The authors have effectively addressed the concerns highlighted.

Reviewer 4 ·

Basic reporting

No additional comments

Experimental design

No additional comments

Validity of the findings

No additional comments

Additional comments

No additional comments